# Urban Niche Assessment: An Approach Integrating Social Media Analysis, Spatial Urban Indicators and Geo-Statistical Techniques

**Iacopo Bernetti** [1,*], **Veronica Alampi Sottini** [1], **Lorenzo Bambi** [2], **Elena Barbierato** [1], **Tommaso Borghini** [2], **Irene Capecchi** [1] **and Claudio Saragosa** [2]

1   Department of Agriculture, Food, Environment and Forestry DAGRI, University of Florence, Piazzale delle Cascine 18, 50144 Firenze, Italy; veronica.alampi@unifi.it (V.A.S.); elena.barbierato@unifi.it (E.B.); Irene.capecchi@unifi.it (I.C.)
2   Department of Architecture DIDA, University of Florence, Via della Mattonaia, 14, 50121 Firenze, Italy; lorenzo.bambi@unifi.it (L.B.); tommaso.borghini@unifi.it (T.B.); claudio.saragosa@unifi.it (C.S.)
*   Correspondence: iacopo.bernetti@unifi.it

**Abstract:** Cities are human ecosystems. Understanding human ecology is important for designing and planning the built environment. The ability to respond to changes and adapt actions in a positive way helps determine the health of cities. Recently, many studies have highlighted the great potential of photographic data shared on the Flickr platform for the analysis of environmental perceptions in landscape and urban planning. Other research works used panoramic images from the Google Street View (GSV) web service to extract urban quality data. Although other researches have used social media to characterize human habitat from an emotional point of view, there is still a lack of knowledge of the correlation between environmental and physical variables of the city and visual perception, especially at a scale suitable for urban planning and design. In ecology, the environmental suitability of a territory for a given biological community is studied through species distribution models (SDM). In this work we have adopted the state of the art of SDM (the ensemble approach) to develop a methodology transferable to cities with different sizes and characteristics that uses data deriving from many sources available on a global scale: social media platform, Google internet services, shared geographical information, remote sensing and geomorphological data. The result of our application in the city of Livorno offers important information on the most significant variables for the conservation, planning and design of urban public spaces at the project scale. However, further research developments will be needed to test the model in cities of different sizes and geographic locations, integrate the model with other social media, other databases and with traditional surveys and improve the quality of indicators that can be derived from information shared on the Internet.

**Keywords:** urban human niche; Flickr; geotagged image; urban metrics; landscape metrics; geo-statistics; urban management

---

## 1. Introduction

### 1.1. General Problem

A growing part of humanity resides in cities that are therefore the typical ecological niche of our biological species [1]. It is a fact that human well-being is influenced by the quality of the city in which he or she lives and walks.

The usefulness of identifying urban spaces with higher visual quality lies in the possibility of searching in these areas specific rules able to replicate positive emotions to be used for the design and regeneration of urban spaces. Following Pallasmaa et al., "Contemporary architecture has often been accused of emotional coldness, restrictive aesthetics and a distance from life. This criticism suggests that we architects have adopted formalist attitudes, instead of tuning our buildings with realities of life and the human mind." [2] (p. 5). The way people use the built environment depends on what the spatial structure created by designers and planners offers. It is a fact that human well-being is influenced by the design quality of the city in which he or she lives. Researchers and multidisciplinary professionals have explored ways to improve the well-being of citizens by changing the built environment. "Urban design may influence people's choices and behavior in the use of the built environment. This influence, however, remains assumption and unclear until urban design qualities can be defined, quantified, measured, and tested empirically" [3] (p. 273).

## 1.2. Literature Review

To understand the interactions between the built environment and the well-being of the population living in the city, Sarkar and Webster [4] proposed the "Urban Health Niche" framework. According to these authors, an individual's health niche is "a spatiotemporal manifestation of the causal agents and processes functioning at the micro-scale, meso-scale and macro scale to produce a specific health state" (p. 34). The built environment shapes and conditions the distribution of social and natural environmental effects that influence the sustainability of the urban health niche. Olsen [5] provides an interesting definition of sustainability in healthcare based on a combination of contextual factors and planning activities. Lifestyle is one of the factors underlying health sustainability, as it determines the burden of the aggregate system in terms of overall costs [6]. In other words, the state of physical and mental health of an individual is conditioned by the environmental variables that define his or her urban health niche. Hence health conditions at both population and individual levels, as they evolve over time, are influenced by urban planning policies and other governance functions on various organizational scales [7]. The built environment, by configuring spaces, facilitates social interactions and therefore can influence the sense of community [8]. Many studies have shown that public green spaces have positive effects on health, promote active life through sport thus reducing obesity [9] and improve mental health [10,11]. A study showed that people who lived in neighborhoods with better quality urban spaces did up to 90 minutes more walking per week than those in neighborhoods with worse "walking habitats" [12]. Walking is by far the most common manner by which the public perceives the quality of an urban environment. Many authors have developed walkability indices to describe urban quality during a walking activity. In a recent review, Wang and Yang [13] emphasized that the main limitation of these approaches is the lack of adequate and accurate data especially for studies that use data on walkability collected through interviews that may be distorted by measurement errors. Although the literature has mainly focused on the study of the correlation between urban spaces and walking activity, comparable results have also been obtained from studies on the identification of bicycling indices [14–16].

In ecology, the environmental suitability of a territory for a given biological community is studied through species distribution models. The species distribution model (SDM), also known as the ecological niche model (ENM), uses geo-statistical models to predict the distribution of a species across geographic space using environmental data. Species distribution models are based on the hypothesis of predicting the spatial potential distribution of a biological species by relating the location of presence to the predictive geographic variables that are supposed to be related to the ecological needs of the species.

In the ecological field, the SDMs are based on the hypothesis of predicting the spatial distribution of a biological species by relating the presence observations with the predictive geographic variables that are presumed to be related to its ecological needs. To transfer these models to the assessment of

the suitability of the urban niche we need observations of locations preferred by the human species in frequenting the open spaces of the city.

In recent years there has been a rapid increase in the use of geo-referenced data extracted from social media to study the relationships between social, physical and economic characteristics of cities and positive and/or negative feelings [17–19]. The rapidly growing number of landscape photographs posted on social networks such as Flickr, Panoramio, Instagram, Twitter or Facebook has great potential for the elicitation of the habitat preferences of individuals. Compared to traditional data sources, such as questionnaires or direct observations, geotagged photographs in crowdsourcing provide an otherwise unavailable perspective on the connections between humans and nature, and facilitate the understanding of preferences regarding the characteristics of the space.

Each social media platform tends to specialize in some characteristics, which makes the choice of crowdsourced media fundamental. In this work we have chosen Flickr because it brings together users with a wide range of motivational factors. Cox [20] discussing the results of a research based on questionnaires, argues that "it makes a lot of sense to analyze Flickr in the context of the social organization of amateur photography as a serious spare time". Furthermore, the photos shared on Flickr are automatically geo-referenced using the coordinates present in the metadata. Each image is then associated with an identifier called Where On Earth ID (WOEID) which identifies in a scaled way the geographic extensions on Earth (from nation to city).

Many studies have highlighted the great potential in photographic data shared on the Flickr platform for the analysis of environmental preferences of people in landscape and urban planning. Research in the city of San Francisco has shown that photographic data shared on the Flickr platform can provide important information for the analysis of environmental preferences in landscape and urban planning [21]. Flickr is an important source of additional information for understanding the perspective of the general public regarding perceptions of an urban environment. Zhou et al. [22] automated the detection of places of interest in multiple cities, based on spatial and temporal features of Flickr images from 2007 forward. Hauthal and Burghardt [23] used Flickr data regarding the city of Dresden (Germany) to extract emotions related to the perception of an urban space, through a sentiment analysis of comments associated with the city. Some research has correlated the geographical density of photos shared by Flickr with the preferences related to urban visual quality detected through questionnaires. Quercia et al. [24] have evaluated the emotional perception of some London paths by correlating the density of photo shooting points shared on Flickr and the emotions declared by over 3000 individuals detected through questionnaires Alampi Sottini et al. [25] in previous research demonstrated the correlation between density of photos shared on Flickr and the perceptive emotional assessments detected through a questionnaire based on differential semantic techniques and administered using a virtual reality headset in the city of Livorno. Finally, in a recent work Yizhuo Li et al. [26] propose a methodological framework to map the global geographical distribution of human emotions extracted through an emotional processing technology from Flickr photos by applying SDM with physical environmental factors.

*1.3. Purpose of The Research*

Although there is a consolidated literature on the use of social media to evaluate the perceived quality of urban habitat, some limits still remain. (1) The geographical scale is generally not adequate to provide useful information for urban management and planning. (2) The indices used in the applications often require specific surveys or data not always accessible globally. (3) Models generally use only one methodological approach chosen a priori (mainly multivariate, maxent or random forest regressions), therefore the stability of the results calculated with different correlation methods is not verified.

Furthermore, some relevant issues for making decisions in urban planning remain unresolved: what are the environmental and physical characteristics important for the quality of the urban habitat from a perceptual point of view? Can we make a ranking of the indices that most influence the visual

quality of the habitat? How much does the marginal variation of an indicator affect the perception of habitat quality?

The strategic objectives of our research are the development of a transferable methodology in cities with different dimensions and characteristics that (a) uses data deriving from many sources available on a global scale: social media platform, Google internet services, shared geographical information, remote sensing and geomorphological data; (b) reduce, through the approach of the ensemble of models, the uncertainty deriving from the choice of the correlation method; (c) offers useful information on the variables significant for the conservation, planning and design of urban public spaces at project level. The results will be further used for the improvement of the existing spaces in cities, and for the design of new open public spaces.

## 2. Materials and Methods

### 2.1. Study Area

Livorno is a municipality in central Italy, spread on 104.5 km$^2$ on the Tuscan (Italy). The perimeter of the study area only includes the urbanized territory of the municipality, as the topic of this study focuses on urban quality. Thus, industrial areas to the north of the city have been excluded, as well as some neighboring areas that are too far from the city center, and those bordering rural areas. Despite the relatively small size, the city of Livorno is composed of a rather heterogeneous mosaic of neighborhoods with unique characteristics due to the different periods of construction. The coastline, with its promenade, also represents a significant geographical macro-characteristic for the perception of the local quality of the urban space. For this reason, it can be an appropriate study area to test a first version of the model to evaluate the quality of urban spaces.

The climate of the city is Mediterranean, with summers mitigated by the sea breeze and non-cold winters. Precipitation is mainly concentrated in spring and autumn. Livorno represents an interesting case study since the climatic well-being indices calculated by processing the weather station data show values particularly suitable for outdoor activities in public urban spaces spread throughout the year: the sunshine index is generally rather high, with up to 7.85 daily average hours of sun; although the heat index is relatively high, with an average of 33.2 annual days with temperature ≥30 °C, the heat wave index is rather low: 18 periods of 3 consecutive days with temperature ≥30 °C; finally there is an ideal summer breeze index, with 6.10 knots of average wind speed.

The urban development of Livorno starts with the plan by Bernardo Buontalenti in the second half of the sixteenth century: until then there was only a small village around a cove. The Buontalenti project for the new city wanted by Medici was characterized by a series of powerful fortifications surrounded by a moat, which gave the city a pentagonal shape (Figure 1, number 1). At the beginning of the seventeenth century a new neighborhood was built crossed by many canals which were therefore called Venezia Nuova (Figure 1, number 2). Subsequently, in 1700 the Grand Duke Pietro Leopoldo built the neighborhoods outside the pentagonal city (Figure 1, number 3). Around the middle of the 19th century, with the development of activities related to the seaside resort, an elegant promenade was created which incorporated the ancient villages of Ardenza and Antignano (Figure 1, number 4). The advent of Fascism coincided with the industrial affirmation of the city which made it necessary to build new neighborhoods for workers. These erroneous urban and architectural models were resumed in the immediate post-war period, when the districts of Sorgenti and Corea (Figure 1, number 5) were built in the emergency due to the shortage of housing (Figure 1, number 6). Meanwhile the historic center, hard hit by the bombings of 1943–44, was partially rebuilt. An improvement in building and urban planning standards took place in the mid-1900s with the creation of modern suburban neighborhoods that definitively welded the city to what were once the external villages of Ardenza and Antignano.

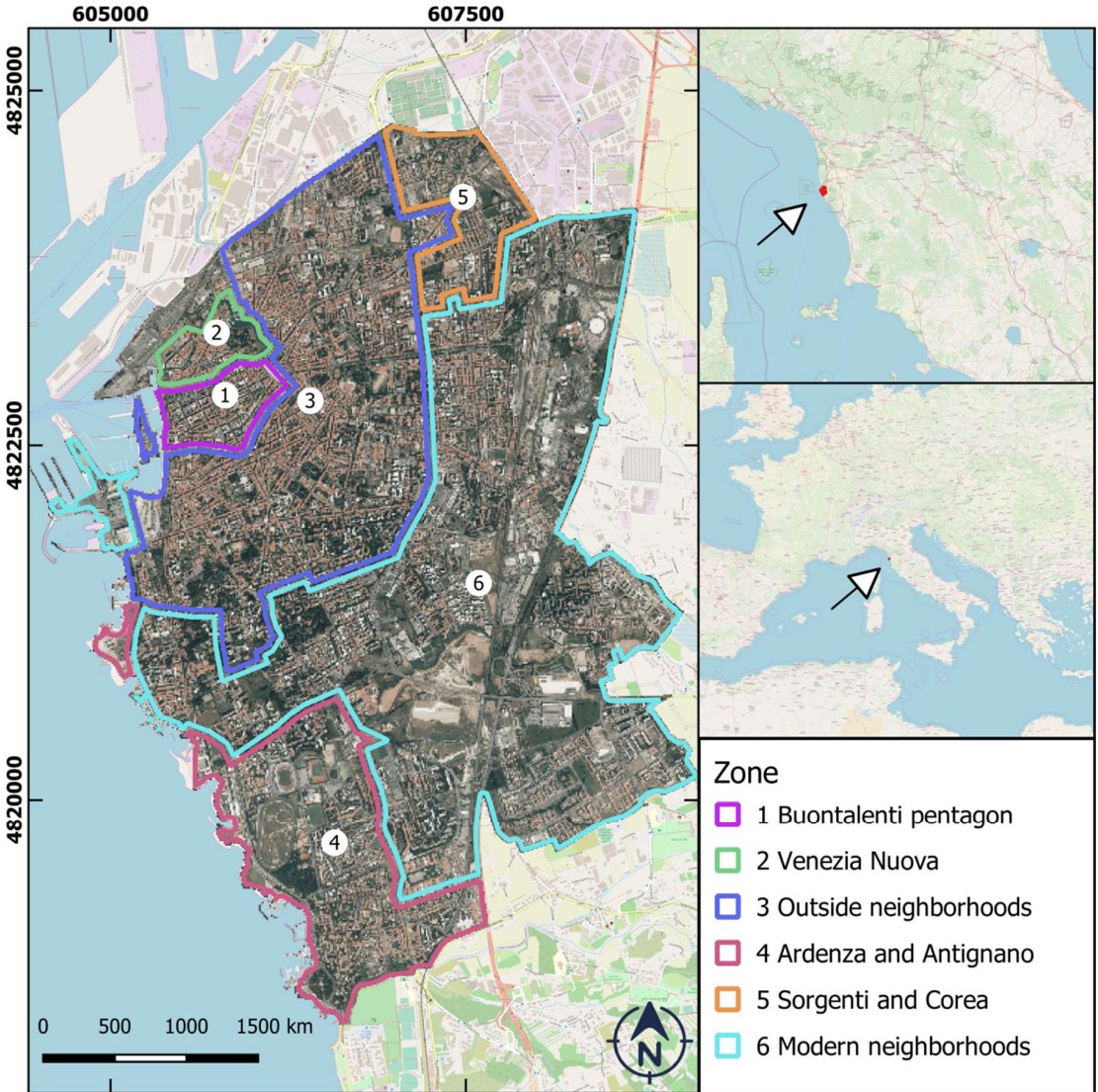

**Figure 1.** Study area.

Livorno is made up of urban areas of different historical periods: 15% of the current buildings were built by 1830, 53% from 1830 to 1954, 22% between 1954 and 1978, 5% between 1978 and 1988 2.5% between 1988 and 1996, 1.5% between 1996 and 2004, and finally 1% between 2004 and 2017. This testifies to the fact that the city has not had significant changes in the local urban fabric in the past 16 years.

Urban green spaces in Livorno cover a total area of 3,133,913 square meters, 13.4 square meters per inhabitant; 55% made up of trees, 27% of hedges and 28% of meadows.

### 2.2. Data Sources

The input data are represented by: (a) remote sensing data; (b) OpenStreetMap map data; (c) geographic digital map data; (d) data downloaded from the Google Street View platform; (e) data downloaded from the Flickr social media. All data refer to a cartographic region defined by the following coordinates: minimum longitude = 10.29321°, minimum latitude = 43.51042°, maximum longitude = 10.35250°, maximum latitude = 43.57019°.

### 2.2.1. Remote Sensing and Geographical Data

The remote sensing data were downloaded from a database of the Tuscan region acquired in October 2013 using an UltraCam digital metric camera Xp (Vexcel) (resolution of 0.2 × 0.2 m). LiDAR data were provided by the Ministry of the Environment, Land and Sea. In this work, the data available from the geographical portal of the Tuscan region with a resolution of 1 × 1 meters was used.

OpenStreetMap (OSM) is a collaborative project aimed at creating free content world maps. The project aims at a worldwide collection of geographic data, with the main purpose of creating maps and cartographies. OpenStreetMap represents the physical characteristics on the ground (roads or buildings) using the tags linked to its basic data structures (its nodes, its ways and its relationships). Each tag describes a geographic attribute of the function shown by that specific node, street or relationship; in this work we used data from the amenity tag.

The cartographic data (roads and coast line) were derived from a regional topographical database (Tuscan region, Italy) with details on a scale of 1: 2000.

### 2.2.2. Google Street View Data

In this study, we downloaded GSV images using the GSV static image API (GSV-API). Each digital photograph (red-green-blue color channel jpeg image) was acquired from the GSV-API at a resolution of 400 by 400 pixels. The images were downloaded via the "Googleway" library, using the statistical software "R." The employed procedure is available as Supplementary Materials. By specifying different parameters in the GSV–API, users can download GSV images with different fields of view (FOVs), heading angles, and pitch angles. In this respect, the heading angle indicates the compass heading of the camera, (heading values range from 0 to 360); the pitch specifies the up or down angle of the camera relative to the street view vehicle, and the field of view determines the horizontal field view of the image. Images from all image collection points along city roads were downloaded every 15 m along each roadway. We set the FOV to 60 and the pitch to 22.5 in relation to the human visual field. Therefore, to cover a 360 ° panoramic view of the surrounding environment, we downloaded 6 images with headings = 0, 60, 120, 180, 240, and 300 for each location.

### 2.2.3. Flickr Data

Using an algorithm based on Flickr's API, the coordinates of shooting points of shared photos from 2005 to 2017 were downloaded. Only photos taken outdoors were selected for analysis. These were achieved by setting the "geo_context parameter" in the API to "outdoors." Moreover, for multiple points with the same coordinates, the same user and the same day were unified. The images were downloaded via the "photosearcher" library, using the statistical software "R." The employed procedure is available as Supplementary Materials.

The raw sample was filtered to eliminate images unrelated to urban quality via filtering tags associated with the photographs. The data extend for a period of about 12 years in order to moderate the effect of annual local variations in photographic density due to the variability of weather conditions (exceptionally hot/cold, rainy/dry, etc.) [27] or to the presence of particular events that could lead to occasional concentrations of shared photos (concerts, events, festivals, etc.). To avoid distortions of perception of the city between night and daytime photos (since the API Flickr gave the possibility to know the shooting date and time) only the photos taken during the day were selected. The sunset time was calculated using the R {suncalc} library.

### 2.3. Methods

The aim of the work is to build an Ecological Model of the Urban Human Niche (EMUHN). The construction phases of an EMUHN are briefly the following:

(a)　identification of the location of individuals' presence in the urban public space;

(b)  identification of the set of environmental variables that determine the habitat model and organization of a geo-database;

(c)  formulation of an ensemble of models based on regression and machine learning approaches for probabilistic prediction of the preferences of the individuals in the urban public spaces and evaluation of the performances of each model.

The flow-chart of the proposed procedure is shown in Figure 2.

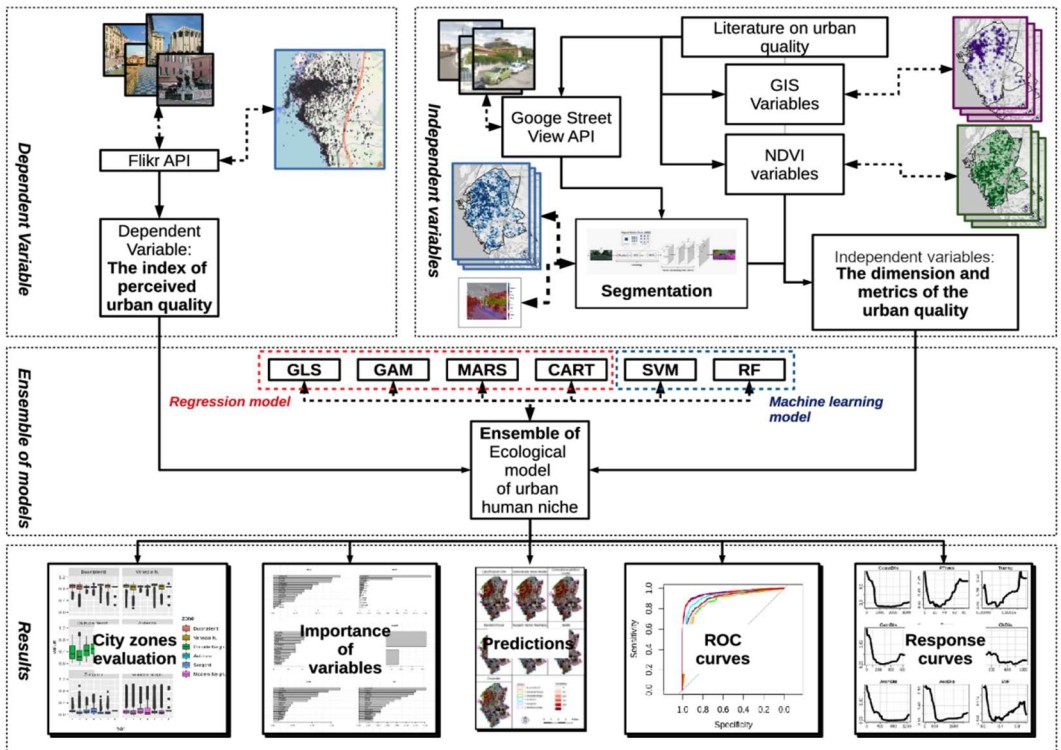

**Figure 2.** Flowchart of the work.

### 2.3.1. Assessment of Urban Niche Indicators

For the identification of the structural and urban variables that condition the use of the urban niche during walking, we have adopted the exhaustive classification of the indicators provided by Ewing and Handy [28]. According to these authors, humans' preference for outdoor space is influenced by the interaction between past experiences, culture and interpretation of the perceived. Urban quality indicators can therefore be classified according to the following typology.

(a)  Enclosure. This category of indicators is characterized by the horizontal and vertical ratios between artificial and natural volumes (buildings and trees) with open spaces (streets and squares). Christopher Alexander et al. [29] (p. 106) stated that "outdoor space is positive when it has a distinct and definite shape, as definite as the shape of a room, and when its shape is as important as the shapes of the buildings which surround it".

(b)  Imageability refers to the quality of a place that makes it distinct, recognizable and memorable. A place has high imageability when physical elements capture attention, evoke positive feelings related to the activity of walking [30].

(c)  Human scale. They are the urban elements that place the individual in relationship with nature.

(d)  Transparency. Transparency is defined as the degree to which people can see or perceive what lies beyond the edge of a road until their gaze reaches the horizon.

(e)  Complexity. Complexity results from varying building shapes, sizes, materials, and colors, but can also refer to diversity in jobs and land use (distribution of housing, offices, roads, etc.).

According to the above, the following quality indicators were selected.

(a) Enclosure: sky view factor; enclosure index.
(b) Imageability: pedestrian index; distance from churches; pavement index.
(c) Human scale: grass density, hedges density, trees density, green index, sidewalk index.
(d) Transparency: transparency index, distance from coast-line.
(e) Complexity: distance from commercials; distance from accommodation land use; distance from buildings with high architectural value.

Most of the indicators selected derive from data available worldwide on the Internet. The remainder derives from geo-databases generally made available by public administrations. The indicators were calculated using the following methodologies.

(i) Deep learning segmentation using GSV imagery was applied to the enclosure index, pedestrian index, cyclist index, road crowdedness index, building crowdedness index, and transparency index.
(ii) Landscape ecology indicators were applied to multispectral remote sensing images for the grass density, hedges density, and trees density.
(iii) The distance operator applied to data from OpenStreetMap database was applied to the urban points of interest (POIs), defined as commercials, accommodations, buildings with high architectural value, and churches.
(iv) The sky view factor was calculated using LIDAR data.

All indices were spatialized on raster maps of the public urban space with a resolution of 1 m × 1 m.

### 2.3.2. Deep Learning Segmentation Using Google Street View Imagery

To detect the indicators of the quality of the urban space we have to detect the targeted physical characteristics in each GSV image [31,32]. In other words, we need to accurately segment the different physical characteristics in each image and label each pixel in the categories to which it belongs. We used a pre-trained network of MATLAB software based on the Deeplabv3+ network. The Deeplabv3+ network is a type of convolutional neural network (CNN) designed for semantic image segmentation [33], with weights initialized by a network trained using the CamVid data-set [34]. It provides pixel-level labels for 11 semantic classes: "Building", "Car", "Cyclist", "Fence", "Pole", "Pavement", "Pedestrian", "Road", "Sky", "SignSymbol" and "Tree".

Based on the theories and recent literature [30–32], we proposed a method to measure four indicators related to the visual quality of urban design, and to calculate the indices for each street site. Table 1 shows the definitions, equations, and explanations of the five indicators.

We used the network-based reverse distance weighting method (NT-IDW) to spatialize the relative values of the sampling points in the raster index maps. The NT-IDW method extends the commonly used spatial interpolation method, IDW (inverse distance weighting), using a network distance instead of the Euclidean distance. NT-IDW was conducted using routines in the ipdw R package.

**Table 1.** Indicators and their definitions, equations, and explanations.

| Indicators | Definition | Equations | Explanation |
|---|---|---|---|
| **Enclosure index** | Degree to which streets and other public spaces are visually defined by vertical elements (buildings, walls, trees) with respect to horizontal elements. | $Encl = \frac{\sum_1^6 Bn + \sum_1^6 Tr}{\sum_1^6 Rd + \sum_1^6 Pv}$ | *Bn* number of building pixels; *Tn* number of tree pixels; Rd number of road pixels; *Pv* number of pavement pixels |
| **Pedestrian and cyclist index** | Degree to which people can see or perceive soft human mobility with respect to unsustainable mobility. | $Ped = \frac{\sum_1^6 Bc + \sum_1^6 Pd}{\sum_1^6 Car}$ | *Bc* number of bicyclist pixels; *Pd* number of pedestrian pixels; and *Car* number of car pixels. |
| **Transparency index** | Degree to which people can see what lies beyond the edge of a street or other public space. | $Trp = \frac{\sum_1^6 Sky}{\sum_1^6 TotPix}$ | *Sky* number of sky pixels; and *TotPix* total pixel number in the image. |
| **Green index** | Extent to which the visibility of street vegetation can influence pedestrian psychological feelings. | $Trp = \frac{\sum_1^6 Veg}{\sum_1^6 TotPix}$ | *Veg* number of vegetation pixels. |
| **Sidewalk index** | Extent to which the visibility of pavement and fences influences pedestrian psychological feelings of surfaces dedicated to walking with respect to surfaces planned for motor vehicles | $Sidewalk = \frac{\sum_1^6 Pav + \sum_1^6 Fenc}{\sum_1^6 Road}$ | *Pav* number of pavement pixels; and *Fenc* number of fences pixels. |

### 2.3.3. Landscape Ecology Indicators Applied to Multi-Spectral Aerial Images

The remote sensing data were used to obtain the coverage and heights of vegetation. The urban vegetation coverage was identified through an analysis of the Normalized Difference Vegetation Index (NDVI) [35]; vegetation was extracted based on NDVI values greater than or equal to 0.2, threshold chosen on the basis of previous research [36,37]. The result was presented as a Boolean map with a resolution of 1 m (similar to LiDAR data), in which the value of 0 indicated the absence of vegetation, and a value of 1 indicated its presence. As urban green areas are characterized by various types of vegetation with different ecologic and perceptive functions, we distinguished these types according to their height values. To obtain the height of the vegetation, we made an overlay operation between the NDVI binary map and a normalized digital surface model generated from LIDAR data. The result of this operation was a raster map divided into three height classes. The first class (from 0 to 0.40 m) represented grass, the second (from 0.40 to 3 m) was classified as hedges, and the third (greater than 3 m) was classified as trees.

The indicators we used were the percentages of green landscapes of class *i* ($P_i$) with *i* = {*grass, hedges, trees*}. The former allowed us to understand the percentage of plant cover of each grid hexagon. The operation is as follows:

$$P_i = \frac{\sum NDVI_{j,i}}{H} \tag{1}$$

In the above, $NDVI_{j,i}$ is the *j*-th pixel in the NDVI raster map classified on class *i*, and *H* is the total hexagon area.

### 2.3.4. Distance From The Urban Points of Interest (POIs)

The locations of urban POIs: commercials, accommodations (hotels, rented rooms resort and the like), churches, and buildings of high architectural value, were derived from information contained in the tag "amenity" of OpenStreetMap database. In an urban environment, the presence of paths limited by public roads does not allow to use the Euclidean distance. We then calculated the distance maps from the POIs using the basic r.cost function of the GRASS software. The algorithm accepts as input a raster map representing the cost surface and a vector layer with the set of points. Each cell in the original cost surface map will contain a category value that represents the crossing cost of that cell. The cells of value 1 in the cost raster are interpreted as road with unit travel cost and the cells of zero value are interpreted as infinite cost.

### 2.3.5. Derivation of Sky-View Factors from LIDAR Data

The sky view factor (SVF) is an extensively used parameter that provides a measure of the degree to which the sky is obscured by the surroundings for a given point. To describe urban climatology at high-resolution scales, several studies have employed the SVF as an urban design quality parameter [38,39]. In this work, the method proposed by Lindberg and Grimmond [40] was used to calculate the SVF from high-resolution urban digital elevation models, using a shadow casting algorithm. The method was applied through Quantum GIS software with the urban multi-scale environmental predictor (UMEP) plug-in [41]. The SVF related to a street canyon of the study area was calculated through a map overlay operation, using the polygonal geo-data of the roads.

### 2.4. Ecological Model of the Urban Human Niche (EMUHN)

### 2.4.1. Generating Pseudo-Absence Data

Ecological niche models aim to estimate the probability that a given localization is suitable for the species examined. All probabilistic algorithms therefore require some form of observation of unsuitable locations, (absence data) which they use as a contrast to the suitable locations detected with the observation of the presence of the species (in our case represented by the shared photo shooting points on Flickr). When absence data is not available, it is necessary to select pseudo-absence data. Several methods have been proposed for this purpose, the choice of which has an important effect on the final results of the model, as highlighted in several previous studies [42,43]. The simplest and most widely applied method for generating pseudo-absences is random selection in the entire study area [44–46], but despite its wide application, the random sampling method increases the risk of introducing false absences into the model in positions that would instead be suitable, leading to underestimate the territorial extension of the habitat [47]. Faced with this problem, it is common practice to set a buffer distance from known places of presence in order to minimize the false negative rate [48]. Since in our application the perceived characteristics of the urban environment depend on the blocks surrounding the individual, the size of the radial exclusion *r* was set on the basis of the average distance between the blocks of the city calculated using the method proposed by Stamp III [49]:

$$r = \sqrt{\frac{S}{0.866 \cdot NB}} \qquad (2)$$

where *S* is the total area of the area, *NB* is the number of blocks.

The values of the raster maps of the urban habitat quality indicators were transferred to the database of points of presence / pseudo-absence through a map-overlay operation.

### 2.4.2. The Ensemble of Models

Ecological niche modelling provides a useful tool for understanding species' distributions as a function of their environmental preferences. However, most studies utilize a single modeling

framework with its specific biases, reducing the comparability of results and potentially limiting predictive capacity; an alternative is to adopt an ensemble ecological niche modeling approach, which combines the output of multiple algorithms into one predictive surface. There are many statistical probabilistic methods that can be used for fitting, selecting and evaluating correlative niche models. The algorithms include regression methods and more complex methods generally based on Machine learning techniques. The following techniques have been used in this work. (a) Regression-based techniques: (a1) Generalized Linear Model (GLM), (a2) Generalized Additive Model (GAM), (a3) Multivariate Adaptive Regression Splines (MARS), (a4) Classification And Regression Tree (CART); (b) machine learning techniques: (b1) Random Forest (RF), (b2) Support Vector Machines (SVM). To obtain a better performance of the models and to avoid over-fitting problems for machine learning techniques, we divided the database into three subsets: a training set (70% of the total observations), used as a set of examples for learning the global model; a validation set (15% of the total observations) for tuning the parameters of the model; and a test set (15% of the total observations), used only to assess the performance of the fully-trained model.

The different statistical techniques have different strengths and weaknesses. The simpler models (regression models) generally have more easily interpretable results and are less subject to over-fitting; on the other hand, they are less efficient from a predictive point of view. The more complex methods (machine learning techniques) do not impose parametric hypotheses and therefore functional forms defined a priori and have better predictive abilities, but they are black-boxes and therefore are difficult to explain.

Comparing the efficiency of models with very different modeling characteristics can be challenging. The significance measures of the estimated parameters are a conventional way to evaluate the performances of a model, but they are relatively easy to apply only for regression and cannot be calculated for ensemble-based methods [50]. As effectively analyzed by Merow et al. [51] the efficiency of a model must be assessed on two aspects: the efficiency of prediction in identifying suitable (urban) habitats and the descriptive capacity of the ecological (urban) niche. One way to compare the models produced by different algorithms is to evaluate the model's predictions on the test data using the Receiver Operating Characteristic curve (ROC curve) method. The ROC curve is graphical patterns for a binary classifier, along the two axes are represented the True Positive Rate (TPR also called "sensitivity") and False Positive Rate (FPR also called "1-specificity"). The ROC curve is created by plotting the value of the TPR, compared to the FPR at various threshold settings. To have a summary index of the efficiency of the classification it is common to calculate the Area Under the Curve ROC (AUC); this indicator ranges between 0 and 1 with 1 corresponds to a perfect classifier and 0.5 is a non-informative classifier.

We also dealt with the evaluation of the models on the basis of the characterization of the ecological niche by analyzing the "response curves" and the "relative importance" of the variables in the model. The univariate "response curves" are commonly used to represent the complexity of the relationships between habitat suitability and single variable and are plotted by calculating the predicted occurrence probability against the predictor of interest keeping all the other predictors at their median values [52]. In other words, the response curves allow the marginal effect of the variation of a variable on the perception of the quality of the urban habitat to be evaluated.

For the calculation of the relative importance of the variables we have adopted the method of the Average Absolute Deviation (AAD) from the median sensitivity calculated on the basis of a one-dimensional sensitivity analysis (1D_SA) [53]. The method adopted is based on the calculation of the results of a model leaving all the variables at the median value and parameterizing the variable a under examination on $L$ levels. For example, if $L = 5$ we have the following parametric levels $x_{aj} = \{0; 0.25; 0: 5; 0.75; 1\}$. For each variable:

$$\{x_a : a \in 1, \{\cdots, M\}\} \tag{3}$$

and for each parametric level $L$ the forecast set is calculated

$$\hat{y}_a = \left\{ \hat{y}_{aj} : J \in 1, \{ \cdots , L\} \right\} \tag{4}$$

For input $\mathbf{x_a}$, the *AAD* measure is:

$$AAD = \sum_{j=i}^{L} \left| \hat{y}_{aj} - \widetilde{y_a} \right| / L \tag{5}$$

where $\widetilde{y_a}$ denote the median of the responses.

### 2.4.3. Ensemble Aggregation Method

To compensate for pros and cons of statistical techniques it is possible to combine model outputs to create a forecast that acquires the advantages of each. Regression-based Combination Methods [54] were used in our work to combine the prediction results of the six models. Using this approach, the ensemble model derives from a linear function of the predictions of the individual models. The weights used in the aggregation are determined using a regression of all the individual predictions on the dependent variable:

$$y = \alpha + \sum_{i=1} \hat{w}_i f_i + \epsilon_i \tag{6}$$

Using a portion of the forecasts to train the regression model, the OLS coefficients can be estimated by way of minimizing the sum of squared errors in Equation (7). The combined model is then given by:

$$e = \hat{\alpha} + \sum_{i=1}^{p} \hat{w}_i f_i \tag{7}$$

with $e$ ensemble predictions, $\alpha$ intercept, $f_i$ predictions of model $i$, $w_i$ combination weight of model $i$.

All models were calculated using R software libraries [55]. The procedures are available as Supplementary Materials.

## 3. Results

### 3.1. Mapping the Indicators of Urban Habitat Quality

#### 3.1.1. Geotagged Photo Distribution

The raw database contained approximately 23,063 photo localizations taken in the period 2005–17, and the final filtered database contained 5796 observations.

Applying Equation (7), an equal number of pseudo-absence points with an exclusion radius r = 100 were generated. Figure 3 shows the distribution of points of presence and pseudo-absence in the areas of the city. Most of the points of presence are located in the Buontalenti (1), Venezia Nuova (2) and Ardenza (4) zones and in some parts of the outside neighbors (3). Sporadic appearances have occurred in Zones 5 (Sorgenti) and 6 (modern neighbors).

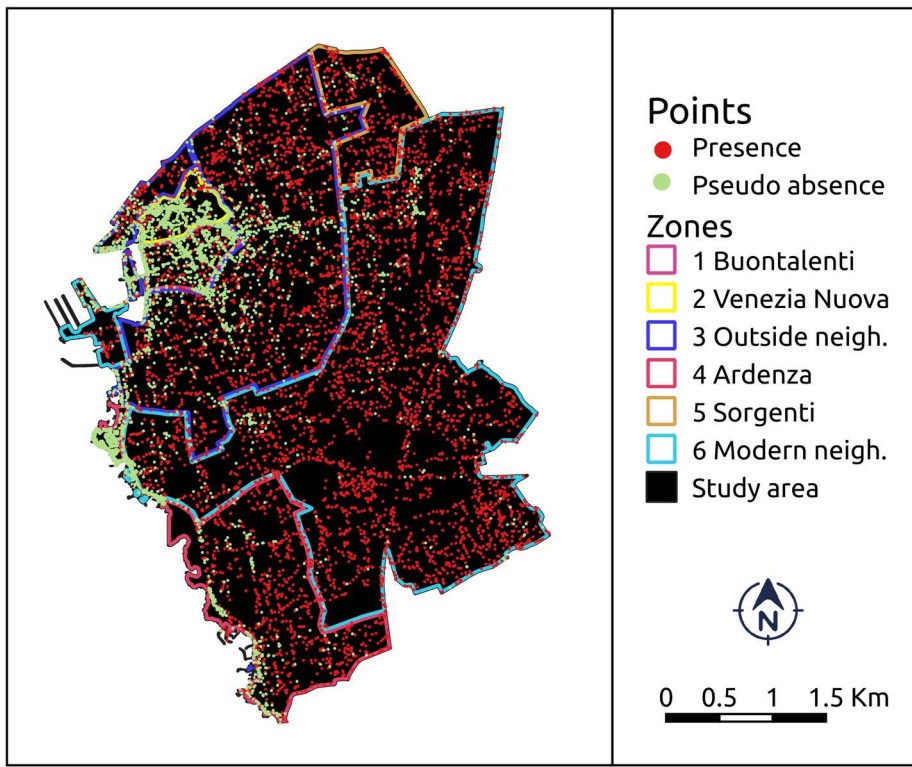

**Figure 3.** Map of the dependent variable.

### 3.1.2. Deep Learning Segmentation Indexes

Using the Googleway library, we downloaded 17,196 geo-tagged images, relating to 2866 sampling points acquired in 2018. Figure 4 shows a typical example of the segmentation process. To validate the network, we extracted 200 random images from the set downloaded with the GSV-API, manually segmented the images, and used them as a validation set for evaluating the performances of the pre-trained network, obtaining an overall accuracy of 89%.

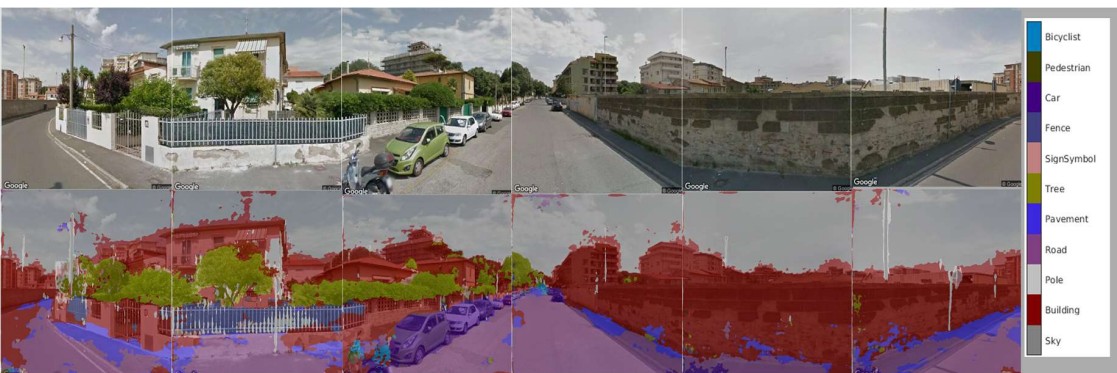

**Figure 4.** Example of segmentation process.

Figure 5 shows the maps of the indices calculated via the segmentation of the GSV images.

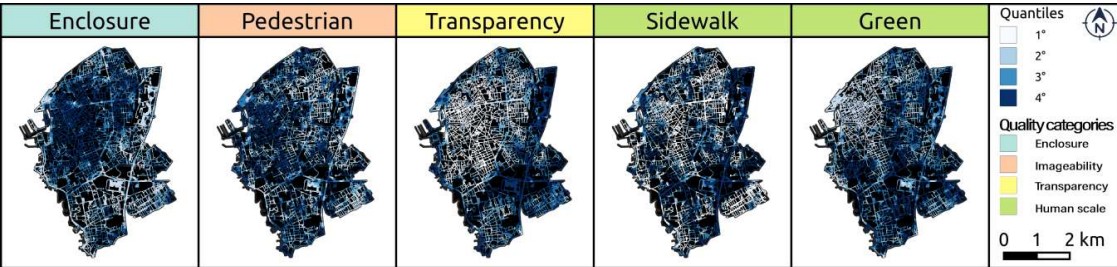

**Figure 5.** Maps of the indices calculated through the segmentation of the Google Street View (GSV) images.

### 3.1.3. Indicators Derived From Geographic and Remote Sensing Data

Figure 6 shows the maps of the three landscape indices. Figure 7 shows the density indices of the buildings intended to influence the perceived quality of the urban environment. Finally, Figure 8 shows the two geographical indices linked to the "visual enclosure" and "transparent" dimensions.

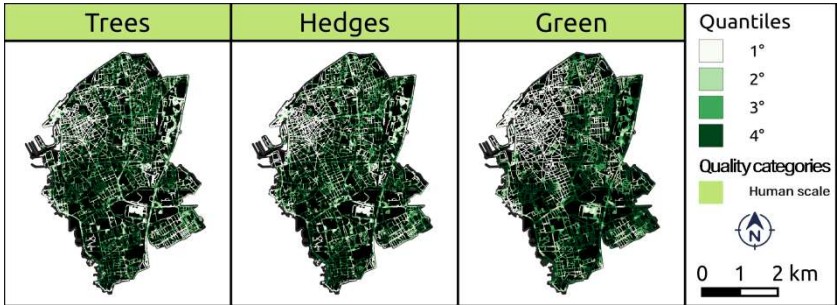

**Figure 6.** Maps of landscape ecology indices.

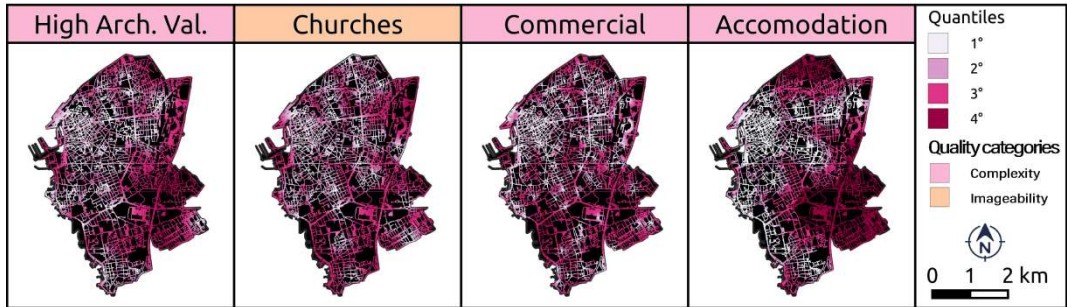

**Figure 7.** Maps of distances from services.

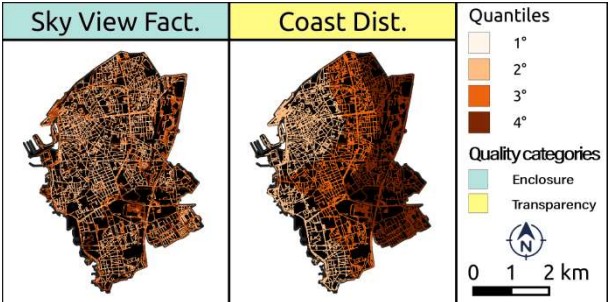

**Figure 8.** Maps of sky view factor (SVF) and of distance from coastline.

### 3.2. Assessment of Features of The Urban Quality

The first step in the EMUHN procedures was testing the multicollinearity between the variables, using Spearman's correlation rank. We kept all of the variables, as they showed a Spearman's correlation lower than 0.7.

#### 3.2.1. Model Evaluation

Figure 9 shows the ROC curves and the AUC values for the six models calculated and for the weighted ensemble of models. The model that has had the best classification performances is the RF model, with an AUC value of 0.956, slightly higher than that obtained by the ensemble of models (0.953). Good classification results are also obtained for the SVM and MARS model, both with AUC values greater than 0.9. The worst predictive performances were those of the CART and GLS models.

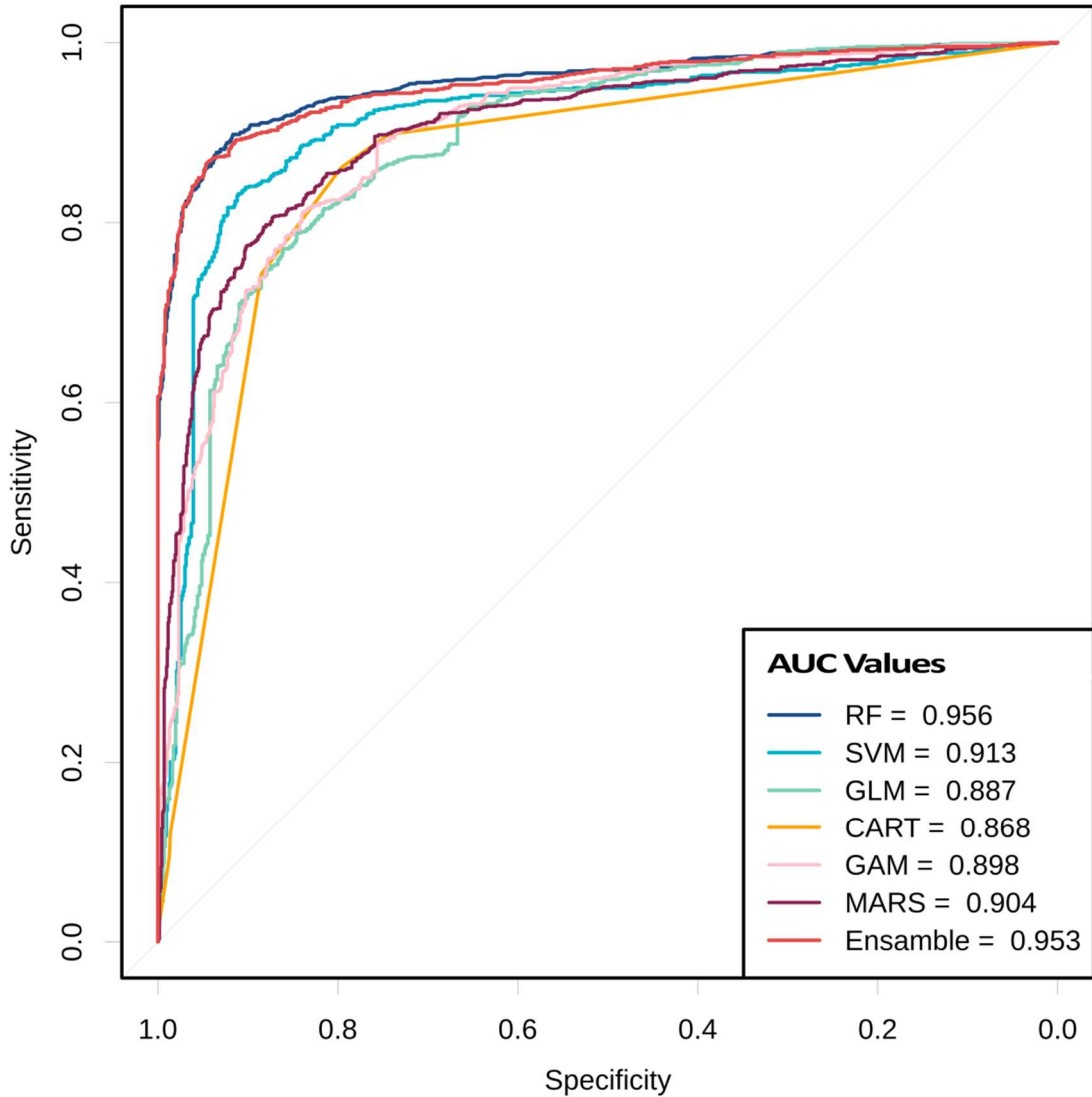

**Figure 9.** ROC curves and AUC values. GLM = Generalized Linear Model, GAM = Generalized Additive Model, MARS = Multivariate Adaptive Regression Splines, CART = Classification And Regression Tree, RF = Random Forest, SVM = Support Vector Machines.

#### 3.2.2. The Importance of Variables

The distance from the coast line is the variable that has shown the greatest relative importance (Figure 10) for all models except the SVM (where it ranks second). The distance from high-value

architectures also has a high relative importance in determining the value of the urban habitat for the GLS, GAM, MARS, CART models. For the remaining variables, the results differ for the two types of methods adopted. The regression methods (GLS, GAM and MARS) have rather similar results; in fact, among the first six variables they all have the distance from the coast, the distance from high-value architectures, the density of hedges and the enclosure index. The machine learning methods, on the other hand, have different results, among the first six variables only the distance from the coast line and the transparency index are common to the two models.

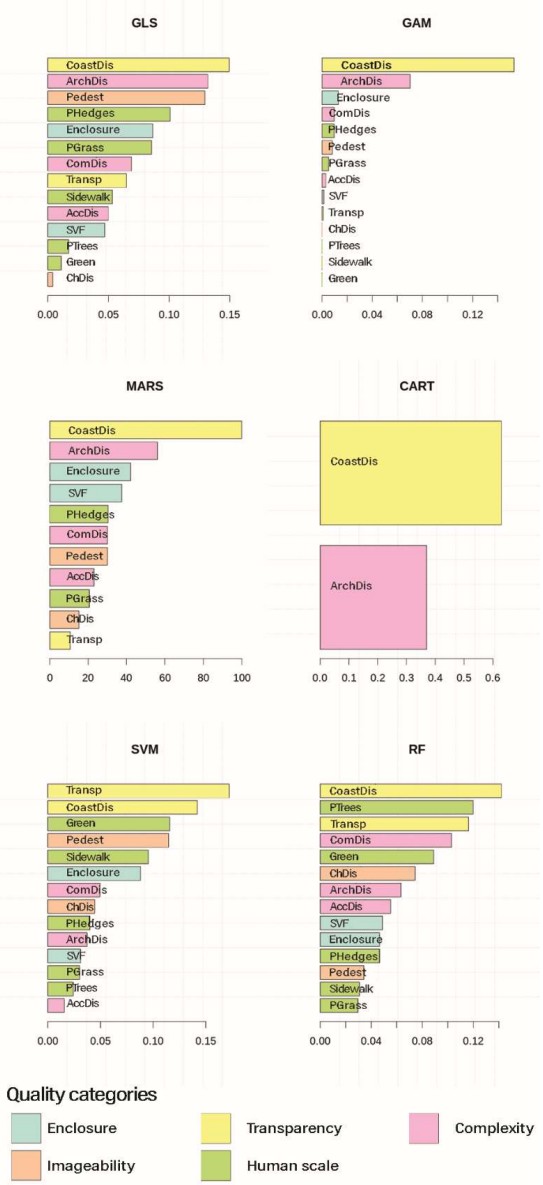

**Figure 10.** Variable importance. SVF = sky view factor, Enclosure = enclosure index, Pedest = pedestrian index, ChDis = church distance, Sidewalk = pavement index, PGrass = grass density, PHedges = hedges density, PTrees = trees density, Green = green index, Sidewalk = sidewalk index, Transp = transparency index, CoastDist = distance from coast line, ComDis = distance from commercial, AccDis = distance from accommodation land use, ArchDis = distance from buildings with high architectural value, ChDis distance from churches. GLM = Generalized Linear Model, GAM = Generalized Additive Model, MARS = Multivariate Adaptive Regression Splines, CART = Classification And Regression Tree, RF = Random Forest, SVM = Support Vector Machines.

With regard to the classification of the indicators in the five categories proposed by Ewing and Handy [28], the methods based on machine learning techniques (RF and SVM) select among the six variables with greater relative importance indicators belonging mainly to the transparency and human scale categories.

Regression methods, on the other hand, have a selection of variables with greater relative importance which is more diversified among the categories. GLS and GAM have in the six most important variables indicators belonging to all four categories. MARS instead selects two variables in the "enclosure" category and two variables in the "complexity" category.

### 3.2.3. The Response Curves

Figure 11 shows the response curves for the six variables with greater relative importance for each model. We evaluated the interpretation of the efficiency of the models on the basis of the criterion of the plausibility of the response curves as a function of consistency with the results obtained from other research that applied direct methods based on questionnaires [28] The GLS model (Figure 11a) has all monotonic curves; the predictors distance from the coast line, distance from high-value architectures, pedestrian index and enclosure index have trends consistent with the results of the cited literature instead the predictors percentage of hedges covers and percentage of grass vegetation do not confirm these results. The GAM model (Figure 11b) has very irregular response curves with generally unrealistic minor fluctuations that cannot be justified for the predictor distance from coastline, distance from the commercial percent of hedges and pedestrian index, while distance from high-value architectures and enclosure index are more realistic and consistent with the state of arts. The curves of the MARS model (Figure 11c) are also monotonic with trends justified by the literature. The CART model (Figure 11d) has very simple and not very explanatory step curves calculated only for completeness of method. Although the RF model (Figure 11e) has the irregular responses typical of classification tree ensemble models, it shows curves that can be interpreted and that are consistent with the results of the research cited. Finally, the SVM model (Figure 11f) instead has more complex response curves; the trend of the curves of the predictors transparence index, distance from coastline, pedestrian index and sidewalk index seem however explainable and consistent with the geographic characteristics of the city, while the green index has an unrealistic trend.

### 3.2.4. The Urban Quality Maps

All the models are comparable to each other in the extension and distribution of the suitability assessment of the urban habitat (Figure 12). However, the CART, RF and SVM models generated more spatially limited forecasts. The GLM and GAM models, on the other hand, have generated wider forecast values and with a tendency to tend to increase throughout the urban area. The ensemble model shows predicted values similar to the RF and SVM models.

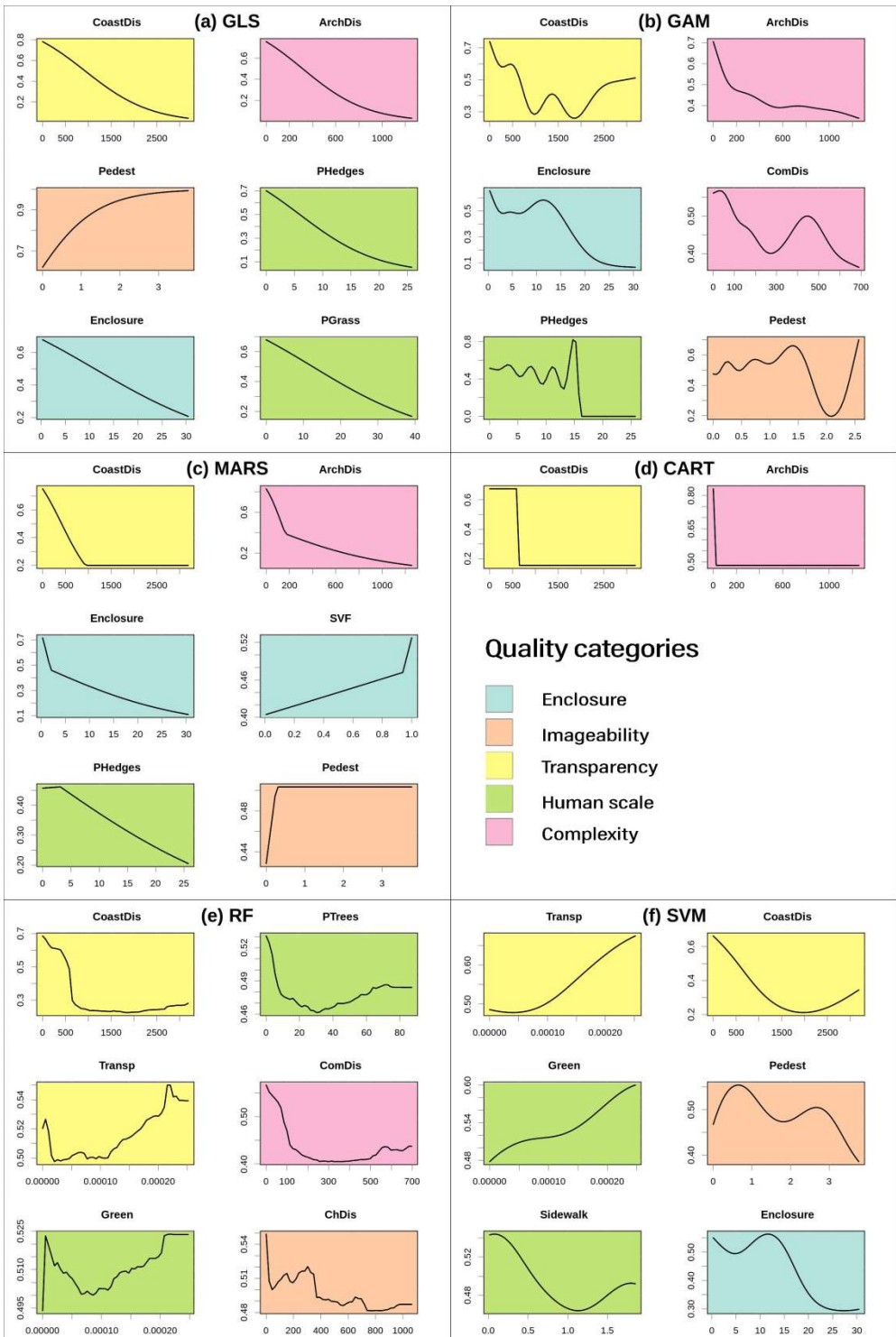

**Figure 11.** Response curves. SVF = sky view factor, Enclosure = enclosure index, Pedest = pedestrian index, ChDis = church distance, Sidewalk = pavement index, PGrass = grass density, PHedges = hedges density, PTrees = trees density, Green = green index, Sidewalk = sidewalk index, Transp = transparency index, CoastDist = distance from coast line, ComDis = distance from commercial, AccDis = distance from accommodation land use, ArchDis = distance from buildings with high architectural value, ChDis distance from churches, GLM = Generalized Linear Model, GAM = Generalized Additive Model, MARS = Multivariate Adaptive Regression Splines, CART = Classification And Regression Tree, RF = Random Forest, SVM = Support Vector Machines.

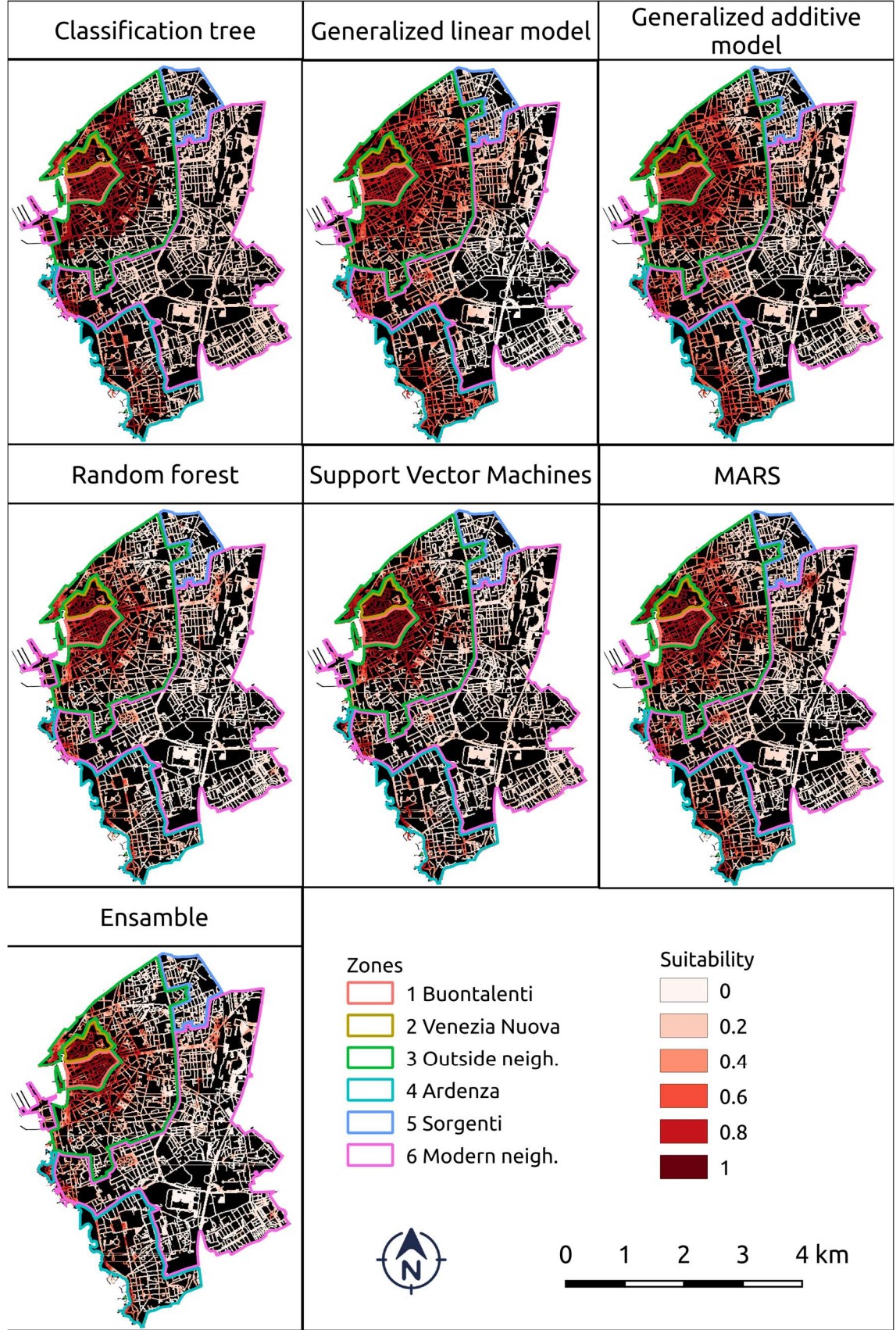

**Figure 12.** Model predictions.

The graphs in Figure 13 allow a better understanding of the locations where there are the greatest discordances and concordances between the models. The zones with the highest suitability index of the urban habitat (Buontalenti and Venezia Nuova) as well as those with the lowest suitability (Sorgenti and Modern neighbors) show similar distributions of frequency of the suitability index between the different models. On the other hand, Ardenza and Outside neighbors have a wider spread of suitability values and greater differences between the models' forecasts.

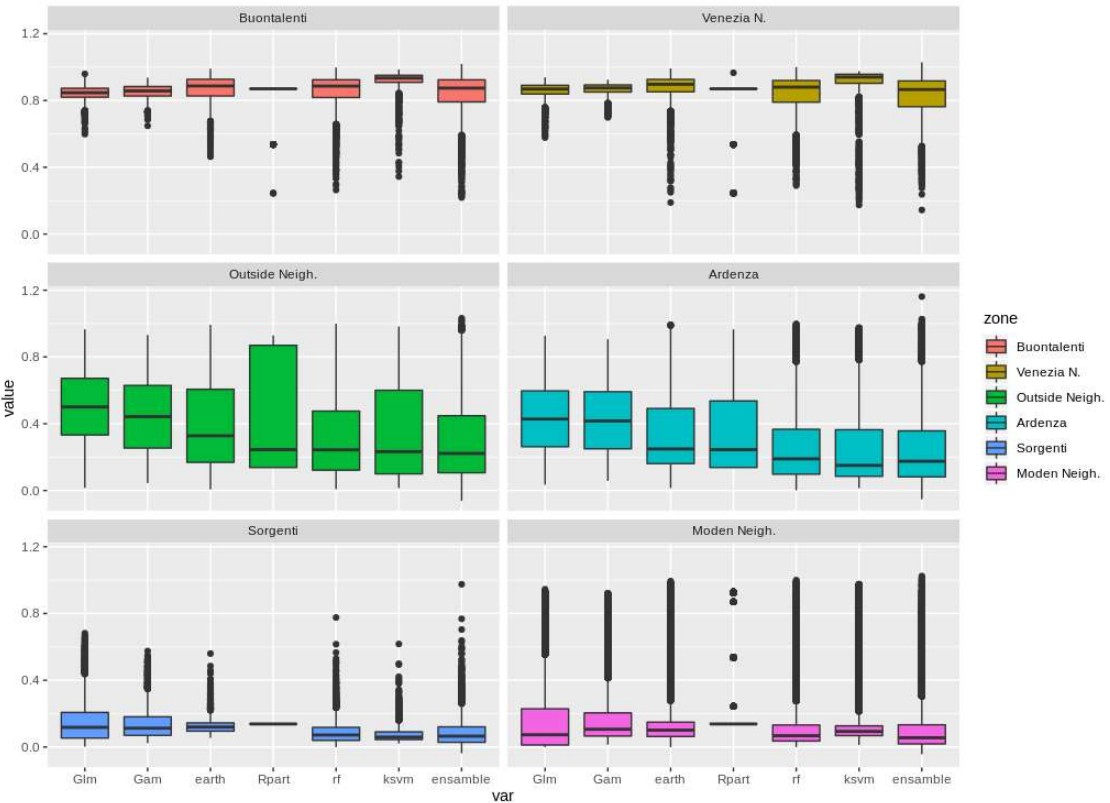

**Figure 13.** Boxplots of model predictions by city zones; glm = Generalized Linear Model, gam = Generalized Additive Model, mars = Multivariate Adaptive Regression Splines, cart = Classification And Regression Tree, rf = Random Forest, ksvm = Support Vector Machines.

## 4. Discussion

Accumulated experiences and knowledge have shown that a well-designed urban habitat can help improve quality of life, state of physical and mental health of an individual [48–50]. Urban planners in European countries are looking for information regarding the urban environment, to facilitate improvements in the happiness of neighborhood residents. This study proposes an integrated method for assessing the human habitat in the urban spaces based on integration between GIS data, remote sensing data, and image segmentation coupled with an ensemble of SDM models.

### 4.1. Have The Research Objectives Been Achieved?

Regarding the first objective of the research the findings showed that indicators calculated on the basis of data of different origins can be integrated through the ecological niche models; in the application we used data deriving from social networks such as Flickr, web services such as GSV, shared geographic information such as OSM, multispectral data from remote sensing and geographic and geomorphological. The model also allows you to combine indices that can be transferred to most cities, with other indicators being specific to the study area. In the case in question, the specific geographical indicator is represented by the distance from the coast line and is specific to coastal

and port. In other cities, geographic indicators of different urban quality could be significant, for example the distance from the waterway for river cities, from the slope for cities characterized by hills. The applicability of the proposed approach in other urban contexts is an important feature of our methodology. Although the indices resulting from the segmentation of GSV images from OSM database and have a global reach, there remains the problem of identifying significant local geographical variables for the perception of the quality of urban spaces. The coast line was important in Livorno, but this specific parameter may not be significant in cities with a cold climate, or a city with an important port. Therefore, for the definition of specific geographical variables for the different urban contexts, important help can derive from the classification of indicators proposed by Ewing and Handy [28]; this classification can be used as a checklist to identify the specific set of variables for the urban contest under study. The "response curves" and "relative importance" methodologies of the variables can instead provide an assessment of the significance of the geographical variables tested.

The second objective of the research was to use the density of the photographs shared in Flickr combined with an ensemble of SDMs to identify the most significant indicators of the perceived urban quality. The use of niche ensemble modeling methods in this study has demonstrated that it is possible to spatialise point observations by identifying the variables that most influence the perception of the urban habitat. Our exploration of the efficiency of the ecological niche model ensembles in the urban context has highlighted the ability of the technique to compare between the predictions of the single algorithm models and to select the models with the best performance on a case-by-case basis. For example, models based on machine learning techniques (RF and SVM) have demonstrated the best predictive capacity while regression models (GLS, GAM, MARS and CART) have been more in concordance in identifying the relative importance of the variables. Although some models have irregular response curves, the stability of the response function (Figure 11) coincided with the selection of the variables; that is, the variables identified as important had more coherent response functions. We also found that the aggregation in ensemble predictions did not necessarily lead to significant improvement in predictive capacity (see Figure 9 and [56]).

Finally, the third objective of the research was to provide useful information for planning and designing public urban spaces. The results of this study can be used in a variety of ways. Municipal planners and administrators can achieve a more detailed and complete understanding of an urban environment. Useful information can derive from the cross-reading of the results obtained by the models. The perceived high-quality environments are concentrated in the historical areas "Buontalenti" and "Venezia Nuova" (Figures 12 and 13) which must therefore be preserved in their most significant characteristics. These characteristics are detectable by analyzing the importance of the variables: quality of the coastline, presence of high-value architectures, enclosure index and transparent index (Figure 10 and Figures 5–8). Response curves can also provide useful information for the design and planning of public spaces. For example, the graphs in Figure 11 show that the effect of the coast on the perception of quality is geographically localized; in fact the curve decays rapidly at a distance of 1–1.5 km from the sea. Similarly, an increase in the distance from churches and buildings of high architectural value is associated with a decrease in the perception of urban quality and the positive effect becomes negligible in less than 1 km. Another interesting result is obtained in the case of the enclosure index: the graph of partial dependence shows that the perception of urban spaces is correlated with low index values, characteristic of the promenade.

### 4.2. Limitation of Study and Topics for Further Researches

Our work clearly has some limitations. In the present paper, we therefore present a prototype of the methodology that will be tested on a small city (Livorno, Italy), but with diversified urban styles and a specific macro-geographical feature. The objective of the paper is to evaluate strengths, weaknesses and further developments of the methodology in order to define a more complex model applicable on a larger scale. The methodology has been tested on a small city and therefore we have been able to download and segment all the images contained in the GSV databases. Processing times

via the ResNet-18 convolutional neural network were therefore acceptable (about 20 sec/image with a core-i7 CPU, for a total of about 100 hours of computation). For larger cities there are two possible solutions that will be implemented in the future development of research. The first consists of tiling the study area; since the methodology is based on a local geographic model this should not bias the estimation of the model parameters. A second solution would be to use a reduced sample of images downloaded from GSV. Pros and cons of the two hypotheses will have to be evaluated. In the future we plan to apply both methods in Florence to verify their advantages and limitations.

Some limitations are also related to the characteristics of social media data. Social media are not just about young people, but it seems that only this social group is actively involved, older people can be mostly recipients of the content and not its creators [26]. Furthermore, the Flickr platform does not allow obtaining social and personal information about the individual user in order to segment visual preferences by age groups and other social variables. It is also not easy to have information on the user's place of origin. In the user profile there is a "city you live in now" field but we have verified that this was declared only in 5.6% of the points used in this work. Previous studies have determined whether users are residents or visitors by looking at the number of photos taken and the length of timestamps on the photos [57]. However, a resident can only take some photos of his city or a visitor can stay in one place for more than a month. With regard to these limitations, we believe that, in the current state of development, the proposed methodology is more suitably usable in synergy with a traditional survey through questionnaires rather than in substitution.

The Flickr social platform allows us to face the problem of urban quality from a perceptual visual point of view, but there are other important socio-demographic elements that define the usability of public spaces [58,59]. A possible future development is to combine our methodology with other approaches based on geo-referenced social platforms, such as Twitter, more oriented towards the communication of moods to analyze the social quality and the perception of security of urban space [60,61].

One aspect we will have to investigate concerns the reasons for taking photographs in a certain place; the reasons can be different: something beautiful is photographed, which creates positive stimuli, but also something bad to provoke negative stimuli of social denunciation. We expect in the future to download through the API the photos taken by Flickr users and to manage them with a supervised trained neural network for the recognition of the emotional content of the images [62].

Another necessary development of the research will concern the time variable. The Flickr images in Livorno are taken throughout the year (21.5% winter, 30.5% spring, 26.7% summer, 21.3% autumn), but GSV images refer only to the month of August. This could lead to a bias due to the growing season of public green areas and further research is therefore needed to define methodologies based on the use of commercial spherical cameras to detect street level images in all seasons of the year. The seasonal factor also affects the climatic variables related to the use of public space (temperatures, rain, wind, etc.); for this purpose we plan to carry out a specific study that correlates the shooting dates of Flickr photos with the daily climatic data of the Copernicus Climate global coverage dataset.

Finally, the CamVid dataset used for training the CNN models is derived from studies on automatic vehicles. Therefore, they have not been trained on the optimal categories for the identification of all of the features that define the quality of an urban space. It will be necessary to further train the CNN models with images classified with semantic categories related to the architectural qualities of roads and buildings. Finally, we plan to include high resolution climatic variables also related to climate change scenarios among the predictors.

## 5. Conclusions

Despite the wide margins for improvement, the proposed approach is able to provide useful information to identify and evaluate the geometric, physical and environmental characteristics of public spaces that most determine the perceived quality of the urban habitat at the planning scale. We believe that our study has helped to demonstrate that the perceived quality of the urban habitat is



influenced by many physical, geometric and environmental variables in a complex way. From a general point of view, data shared by social media combined with data deriving from Google's Internet services, shared geographical information and remote sensing via SDM can provide a useful tool to improve our understanding of the relationship between humans and the urban environment, and help develop targeted and efficient management practices.

**Supplementary Materials:** The following are available online at http://www.mdpi.com/2071-1050/12/10/3982/s1, StreetViewHarvester.R: Cran R procedure. MATLAB_procedure.m: MATLAB procedure.

**Author Contributions:** Conceptualization, I.B., E.B., I.C. and T.B.; methodology, I.B., L.B., E.B. and T.B.; software, L.B., T.B.; data curation, I.B., E.B.; validation, I.C., C.S.; formal analysis, I.C.; data curation, I.B., E.B.; writing—original draft preparation, I.B., E.B., V.A.S.; visualization, E.B., I.C.; supervision, C.S., I.B.; funding acquisition, C.S. All authors have read and agreed to the published version of the manuscript.

**Funding:** This research received no external funding.

**Conflicts of Interest:** The authors declare no conflicts of interest.

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
