# Peer review of "Urban Niche Assessment: An Approach Integrating Social Media Analysis, Spatial Urban Indicators and Geo-Statistical Techniques"

_sustainability, doi:10.3390/su12103982_

Round 1

Reviewer 1 Report

The authors sought to characterize suitability assessment of urban habitats by applying theoretical species distribution models with online available image data repositories and machine learning modeling. Using social media (Flikr), panoramas from online APIs (Google Street View) and machine learning methods were applied to identify features that influence perceived quality of urban environments. These methods can be useful for city designing, visitation advertising, and further development of methods that integrate online available data and image processing for quantifying geo-social sentiments that can be difficult to quantify using traditional survey methods.

Strengths:

The methods and theoretical framework were well detailed and outlined with supportive figures to illustrate geospatial main findings and model comparisons.
The longitudinal sampling (~12 years of data) and standardizing the final data set (i.e., filtering time of day) helps to strengthen the findings.
The detailed model comparison promotes robustness of the statistical analysis.

Weaknesses:

This project focused on a single Italian city. Will this methodology scale up? Authors provide some thoughts (lines 582-586) regarding how these methods can scale to larger cities, either by tiling the study areas or using a sampling of images downloaded.

The project focused on data from the Flikr social media platform. Although, this platform was very relevant for the presented analysis, could other social media platforms provide similar functionality? The thought process behind choosing Flikr over other social media platforms was appropriately outlined in the methods (lines 156-184). Perhaps other implementations of social media repositories, for example geo coordinates associated with Twitter or Instagram data, be similarly applied to identify regions of population density/interest and then applying their methods for capturing panoramas using Google Street View. A brief discussion of potential implementations of their methodology using these other social media platforms may help strengthen the discussion and future applications of this research project.

Other comments:

On figure 9 the color choices for each model ROC curve are aesthetically challenging (i.e, the bright colors on the white background are not visually pleasing). Also, a figure title, larger tick fonts and text label fonts may improve readability.

Reviewer 2 Report

The authors presented a very good geostatistical workshop and geodata analysis, the selection of geostatistical methods, and the quality of research results. In view of this part of the work I have not found any shortcomings or doubts.

The authors presented an interesting concept of marking pseudo presence and pseudo absence in designated public spaces. The statement that the existence of a large number of photographs placed in social media in a given location in the city is not overestimated. Certainly, there is a strong correlation between the emotions occurring during the occurrence of visual stimuli. This results in a desire to share one's impressions with the public, which makes places more interesting than others more often geotagged.

There could only be a doubt that the authors are rightly and comprehensively trying to solve the issues of urban indicators. In the publication, the authors have dealt very well with the "normalization" of the acquired photographs, by applying filtration and rejecting a significant number of objects that are not useful for research work. As far as the selection of research material related to the same conditions of taking photographs is concerned, I have no comments, the authors have taken care of their optimal and objective parameters. In my opinion, we should learn a little more about the reasons for taking photographs in a given place. The reasons may be different: something nice is photographed, which creates positive stimuli. You also take pictures of something ugly - negative stimuli. Another reason may be the presence of a different style of buildings, streets or other objects than in the place where the Flickr user comes from. Please comment on this in the text - preferably in the discussion on this issue.

Social media is not only about young people, but it seems that only this group engages with a strong, interactive approach. Older people can be mostly recipients of the content and not its creators. In view of this assumption, the introduction to this paper, as well as the discussion, should briefly discuss how the diversity of authors of photographs on Flickr in terms of age could affect the results of the research. It would also need to be answered whether the authors find it possible to obtain anonymous data about the authors of the photographs and whether such actions are envisaged in the future.

Another issue is whether the photographs were taken by people living in the research area (city) or by visitors. The text of the paper lists one parameter marked as "distance from accommodation", which suggests that the authors of the publication could have such data. How was the distance from the accommodation examined?

The parameter "Imageability" is not very clearly defined, as "It is the quality of a physical environment that evokes a positive and reassuring experience for the observer: Lynch ([22], p. 9) defines imageability as the "... shape, color or arrangement which facilitates the making of vividly identified powerfully structured, highly useful mental images of the environment". And then by referring to the definition of the Imageability indicator in this form of description: "pedestrian index; distance from churches; pavement index". In my opinion these forms of description are incompatible and have a different character.

In the introduction it was written that " The strategic goal of our research is to develop a methodology based on SDM approach considering the city as an ecological niche of the human species.", but I think that this is just a concept of methodology, especially as the publication presents in depth, but nevertheless general assumptions and discussion of results. If the current wording is maintained, the proposal of your methodology should be confronted with other approaches. Please consider changing the wording.

There is a clerical error in line 192: trees and not tress.

Reviewer 3 Report

Comments to the Author:

The reviewer read this article with great interest. The purpose of the research was to capitalize on geosocial media data and regression models to help understand how to improve urban spaces. This was an interesting study, as it married new data sources (i.e., Flickr) with regression, species distribution models, and machine learning algorithms to highlight preferences of individuals for urban spaces . While this reviewer was initially very excited to review this manuscript, it soon waned due to the articles poor structure, lack of clear research objectives, and unorganized/weak literature review. The main issue with the paper is there doesn’t seem to be a literature contribution here. There are some research “objectives” throughout the paper, but they are all in contrast and not in the expected places. This isn’t helped by the thin and unorganized literature review; coupled with numerous grammatical errors throughout the paper. Moreover, without a well- designed manuscript and established research objectives, this paper appears to be merely a utilitarian exercise on using Flickr data and common regression models. The detailed comments and questions are below.

  1. It is recommended that the author thoroughly review the manuscript for all grammatical and punctuation errors. They are too numerous to count.

  1. Abstract needs a major revision. It doesn’t answer the most basic question about the research: “so what?” Besides introducing the reader to the topic, it should provide significant results and the validity of the research; this would be useful for stakeholders.  Geostatistical methods is mentioned, but is nowhere to be found in the paper? The results are not novel according to this abstract. For instance, many past works have used geosocial media to characterize urban habitat. See: Yizhuo Li, Teng Fei, Yingjing Huang, Jun Li, Xiang Li, Fan Zhang, Yuhao Kang & Guofeng Wu (2020): Emotional habitat: mapping the global geographic distribution of human emotion with physical environmental factors using a species distribution model, International Journal of Geographical Information Science, DOI:10.1080/13658816.2020.1755040; Rybarczyk, G., Banerjee, S., Starking-Szymanski, M. D., & Shaker, R. R. (2018). Travel and us: The impact of mode share on sentiment using geo-social media and GIS. Journal of Location Based Services, 12(1), 40-62; Huang, Q., & Wong, D. W. (2016). Activity patterns, socioeconomic status and urban spatial structure: what can social media data tell us? International Journal of Geographical Information Science, 30(9), 1873-1898; Li, L., Goodchild, M. F., & Xu, B. (2013). Spatial, temporal, and socioeconomic patterns in the use of Twitter and Flickr. Cartography and Geographic Information Science, 40(2), 61-77.

  1. Introduction; it needs a thorough revision. The revision does not develop a foundation of knowledge highlighting the purpose of this study. The Introduction should follow this format:
    1. Broad topic: general problem and background, motivation (why we should care), reference lit
    2. Narrower topic: zero in on specific problem/issue of this paper, including previous research in the area (both geographic and topical as relevant) and definition of key concepts and assumptions; reference lit
    3. Conclude with a paragraph that clearly states the research question, purpose of this research.

There are also some interesting sentences in this section that need clarification. Lines 56-57 on walkability. What’s the purpose of speaking of walkability indices? Why not bicycling indices? I think the authors mean to elaborate on the problems with typical qualitative data collection methods, and if so, needs to be concentrated on and supported by past works. See these works for more information on this:

Salon, D. (2016). Estimating pedestrian and cyclist activity at the neighborhood scale. Journal of Transport Geography, 55, 11-21; Neutens, T., T. Schwanen. (2011). The prism of everyday life: Towards a new research agenda for time geography. Transport Reviews, 31(1), 25-47.

                There is a major grammar error with lines 62-63.

Lines 71-76 appear to the paper’s main thesis/research objective. As mentioned earlier, this is weak and needs to be expanded. The research questions/objectives need to be stated early and must remain consistent, as well as referenced, throughout the paper. Here is a guide:

  1. Should be explicitly spelled out (e.g., in the form of a question)
  2. Must be operationalized from general concepts (sustainability, food security) into specific, observable, measurable variables (paper recycling rates, biofuel usage, etc.)

A separate literature review section is needed to allow the reader to understand past research on the use of geosocial media; issues surrounding qualitative data collection; and use of species distribution models. This will help everyone understand why this research was undertaken.

  1. Materials and Methods. The expected format has not been followed. It should adhere to these basic tenants:
    1. Someone should be able replicate your study by reading this. Needs to answer who, what, when, where, why and how. (ca. 800 words)
    2. Case description and selection
      1. Subjects/case used
      2. Description of field site, relevant physical and biological features, map
      3. How were participants selected and/or recruited?
      4. Why is this an interesting case- why study this phenomenon in this particular context?
      5. Origins of samples, data, and materials

Section 2.2.1. The first four paragraphs could easily be transferred to a literature review. These paragraphs have no reason to be included in a methods section (see a and b above).

Line 178 is grammatically incorrect.

Section 2.2.1., line 215. What is GSV? The reader at this point does not know this acronym.

  1. Results. This section is generally ok but very hard to follow. It is recommended that it be simplified and reorganized by the research questions using sub-headings.

  1. Discussion. This section is a good start only. After reading this section twice, it is highly recommended that it get a major revision. It doesn’t seem aligned with the Abstract or Introduction section. The guide to follow on what a results and discussion should consist of are noted below.
    1. Discussion section:
      1. Re-state research question
      2. Interpret findings from Results in light of this question
  • Do results agree with what others have shown? (go from narrow back to broad by linking your specific findings with theoretical context from Intro)
  1. Reflections on this study
    1. Possible sources of error
    2. How could study be improved?
    3. What would next steps be (in this study, in this field)?
    4. What are the implications?

Line 529 is suspect. This reviewer is certain that past researchers have investigated the links between social services and the environment. This statement isn’t aligned with the main focus of the paper or literature, so it is perplexing.

Lines 530-532 are the first time that any research objectives are announced. Then line 539 only refers back to objective (b), but is different than the one indicated in line 531? This is highly confusing.

The last two paragraphs (lines 553-564) have the correct tone, but this reviewer is attempting to understand the purpose of this study other than an utilitarian exercise?

  1. Conclusion. This section needs to be revised for content and grammar. The first paragraph should essentially deliver a similar message to your title and the last part of your abstract but in different phrasing. What does your work mean in the big picture? What is your key takeaway message? These answers are key to an acceptable conclusion section.

Line 578 – grammatically incorrect

Reviewer 4 Report

This paper describes a prototype process for evaluating the usability of urban spaces for human activity based on physical structure and aesthetic. The authors use a social media and Google Street View images, several human niche indicators, and a suite of quantitative methods and tools to map areas with high usability in Livorno, Italy.

This paper and method present a viable and compelling strategy for mapping variability in urban usability and would make a good contribution to the literature. I have a few broad comments which would improve the paper, its scope and contribution interdisciplinarily, and the project as it moves to broader application in more cities.

The introduction would benefit from a clearer thesis statement. The thesis should make clear what the project intends to do. The paragraph consisting of lines 70-76 are the closest to the paper’s the thesis statement that I see, but it also contains things that are not further covered in the paper.

Since the authors suggest this project is a prototype for a larger application in other urban contexts (lines 70-76 emphasize this), I think this paper would benefit from thinking more explicitly about how the limitations of the one geographic context of Livorno would influence the application of these methods in other places and times. Similarly, lines 70-76 suggest a further application of these methods “for the improvement of the existing spaces in cities, and for the design of new open public spaces.” This is never discussed elsewhere as an outcome, and arguably, the indicators that you use already say what needs to be done to public spaces to make them more usable, so how does your model contribute to that goal? The conclusion should address this more extensively.

The application of this model in other contexts is also important. In a positive way, the authors discuss the meaning of ‘distance to coast’ as a factor in the model if applied to a city without a coast but has some other attractive feature like waterways and hills.. But how can you measure the relative significance of the coast to Livorno residents to the significance of the coast to residents of a city with a cold climate, or a city with a major port, or a city where a large urban park is more attractive than the coast, or to the hill in a city without a coast? The variable is very different depending on the context.

Similarly, the people and spatial organization of cities have a lot to do with the use of the space. For instance, American cities are highly segregated by race, and use of public space is correlated with sociodemographic traits, more than the appearance of the space. I suggest perusing some of the literature on public space and sociological factors. They would suggest that just changing the nature of public spaces is not enough to make them more usable; there is a lot more factors besides what is in your model to public space usability. Here are 2 examples, both from Progress in Human Geography, which engages a lot with public space:

(Byrne and Wolch, Nature, race, and parks: past research and future directions for geographic research) https://journals.sagepub.com/doi/abs/10.1177/0309132509103156

(Ye, Re-orienting geographies of urban diversity and coexistence: Analyzing inclusion and difference in public space) https://journals.sagepub.com/doi/abs/10.1177/0309132518768405

I think the authors should consider the variable of time more critically. The static images were taken at specific times of day and year when activity is certainly going to be more variable. GSV images are more likely to be taken during normal working hours. What if images were taken during low vegetation seasons? An honest explanation of how time will impact the image contents and the model results is necessary.

Some model questions:

What is the ‘Pol’ in the enclosure index formula? This is not described in Table 1

Some of the denominators in the equations are confusing to me. Theoretically, they are ‘detractors’ from what he index is measuring, but why are number of car pixels a denominator for human activity? And total pixels is used as a denominator in the vegetation index but no other indicator?

Typo:

Figure 2 ‘indipendent’

Round 2

Reviewer 3 Report

After re-reviewing the manuscript it is clear that the authors capitalized on all reviewer recommendations. Provided this, the article has been improved upon in three key areas: style/organization; literature review; and discussion of results. In particular, a more robust explanation of how the results make an impact on the literature were noted. The improvements to the conclusion section are also noteworthy. Including verbiage on the applications, limitations, and future work was something that the first draft sorely lacked.

These key additions have now brought this manuscript into publishable form.